

# Ultra-distal tephra deposits and Bayesian modelling constrain a variable marine radiocarbon offset in Placentia Bay, Newfoundland

Alistair J. Monteath[1], Matthew S. M. Bolton[2], Jordan Harvey[2], Marit-Solveig Seidenkrantz[3], Christof Pearce[3], Britta Jensen[2]

[1]Geography and Environmental Science, University of Southampton, Southampton, SO17 1BJ, UK
[2]Earth and Atmospheric Sciences, University of Alberta, Edmonton, T6G 2E3, Canada
[3]Department of Geoscience, Arctic Research Centre, and iClimate, Aarhus University, Aarhus, DK-8000, Denmark

*Correspondence to*: Alistair J. Monteath (a.j.monteath@soton.ac.uk) or Matthew Bolton (bolton1@ualberta.ca)

**Abstract.** Radiocarbon dating marine sediments is complicated by the strongly heterogeneous age of ocean waters.
Tephrochronology provides a well established method to constrain the age of local radiocarbon reservoirs and more accurately calibrate dates. Numerous ultra-distal cryptotephra deposits (non-visible volcanic ash >3000 km from source) have been identified in peatlands and lake sediments across north-eastern North America, and correlated with volcanic arcs in the Pacific north-west. Previously, however, these isochrons have never been identified in sediments from the north-west Atlantic Ocean. In this study, we report the presence of two ultra-distal cryptotephra deposits; Mazama Ash and White River
Ash eastern lobe (WRAe), in Placentia Bay, North Atlantic Ocean. We use these well dated isochrons to constrain the local marine radiocarbon reservoir offset (ΔR) and develop a robust Bayesian age-depth model with a ΔR that varies through time. Our results indicate that the marine radiocarbon offset in Placentia Bay was -126±151 years (relative to the Marine20 calbration curve) at the time of Mazama Ash deposition (7622±18 C.E.) and -396±144 years at the time of WRAe deposition (852-853 C.E.). Changes in ΔR coincide with inferred shifts in relative influences of the Labrador Current and the
Slopewater curret in the bay. An important conclusion is that single-offset models of ΔR are easiest to apply and often hard to disprove. However, such models may oversimplify reservoir effects in a core, even over relatively short time scales. Acknowledging potentially varying offsets is critical when ocean circulation and ventilation characteristics have differed over time. The addition of tephra isochrons permits the calculation of semi-independent reservoir corrections and verification of the single ΔR model.

**1 Introduction**

Tephrochronology (the "*identification, correlation and dating of tephra layers*"; Thorarinsson, 1981; Lowe and Hunt, 2001) is an age-equivalent technique that can be used to date or synchronise, palaeoenvironmental and archaeological records over a range of timescales and distances (Lowe, 2011). This method is particularly valuable in establishing chronologies for marine sediment records that are inevitably depleted in radiocarbon relative to the atmosphere. The depletion in radiocarbon
is mainly a result of long oceanic residence times and is called the marine reservoir age (R) (Reimer and Reimer, 2001). Due



to large stocks of "old carbon" in ocean waters, organisms that incorporate marine carbon (e.g. foraminifera, fish, marine mammals, molluscs, etc.) typically have a radiocarbon age that appears older than terrestrial organisms of an equivalent age (Ascough et al., 2005). Therefore, a correction must be applied to accurately calibrate radiocarbon dates from ocean sediments (Heaton et al., 2020). Selecting an appropriate correction, however, is not straightforward as the radiocarbon age

of ocean water bodies is strongly heterogeneous and may not have remained consistent through time (Gordon and Harkness, 1992; Reimer and Reimer, 2001; Alves et al., 2018). Deviations from the average marine reservoir age are expressed as the local marine radiocarbon reservoir offset (ΔR). Tephrochronology provides an independent means to partially address these issues and establish local marine radiocarbon offsets (e.g., Pearce et al., 2017). These data can also reveal past changes in ocean circulation. For example, a comparison between age-depth models established from radiocarbon-dated marine

macrofossils (e.g., molluscs and foraminifera) and tephrochronology showed that local marine radiocarbon offsets on the north Icelandic Shelf varied by up to 450 years as the influence of radiocarbon-depleted, Arctic water masses fluctuated (Knudsen and Eiríksson, 2002; Eiríksson et al., 2004, 2011).

Since the identification of ultra-distal cryptotephra deposits in Nordan's Pond Bog, Newfoundland (Pyne-O'Donnell et al., 2012) (Fig. 1), a series of studies have investigated peatlands and lake deposits, throughout the eastern seaboard of north-

eastern North America, for the presence of volcanic ash (e.g. Jensen et al., 2014; Pyne-O'Donnell et al., 2016; Mackay et al., 2016, 2022; Spano et al., 2017; Monteath et al., 2019). This research has identified >30 well defined cryptotephra deposits (Jensen et al., 2021), some of which extend into the North Atlantic region (Zdanowicz et al., 1999; Jennings et al., 2014) and as far as western Europe (Jensen et al., 2014; Plunket and Pilcher, 2018). These tephra deposits are derived from a range of eruption sizes (e.g., Mazama Ash ~176 km$^3$ erupted volume, Buckland et al., 2020; South Mono 0.171-0.195 km$^3$ erupted

volume, Bursik et al., 2014), where long-range deposition was likely influenced by some combination of eruption size, style, duration, and atmospheric conditions/circulation (e.g., the jet stream; Bursik et al. 2009). While large explosive eruptions may be expected to affect greater areas, in general, understanding what is controlling the exceptional dispersal of some tephra deposits and not others is still not well resolved (Pyne-O'Donnell et al., 2012; Jensen et al., 2021).

The Holocene cryptotephra record is uniquely well dated, through a network of chronometers and age models, including

layer counting in ice cores (Sigl et al., 2016, 2022; Toohey and Sigl, 2017). There has, however, been no successful attempt to extend eastern North America's tephrostratigraphic framework to ocean cores in the north-western North Atlantic Ocean. Resolving chronological ambiguity in palaeoceanographic records from this region would be particularly valuable as it includes the confluence of the Labrador Current and the North Atlantic Current – both of which are influential components of the sub-polar gyre and Atlantic Meridional Overturning Circulation (AMOC) (Fig. 1). In this study, we identify ultra-

distal, North American cryptotephra deposits in marine gravity core AI07-10G from Placentia Bay at the western seaboard of the North Atlantic Ocean (Fig. 1). We go on to use these isochrons to constrain the local marine radiocarbon reservoir offset and develop a robust Bayesian age-depth model. Finally, we highlight the potential for further studies of North American, ultra-distal cryptotephra deposits in ocean sediments while considering some remaining challenges.



## 1.1 Placentia Bay, North Atlantic Ocean

Placentia Bay is located immediately south of Newfoundland, Canada, on the north-western margin of the North Atlantic Ocean. The bay is bordered by the Avalon Peninsula to the east, the Burin Peninsula to the west and the Isthmus of Avalon to the North (Fig. 1). To the south, the seaward opening of the bay is approximately 100 km wide. Water depths exceed 400 m in the bay, which is around 130 km long. Placentia Bay is typically free from sea ice year round, although ice can form between mid-February and late April during the coldest winters. Iceberg sightings in the bay are rare. Between C.E. 1974-

2003 sightings only occurred in seven years (30 sightings total) (Catto et al., 1999; Mello and Rose, 2005). The hydrology of the bay is strongly influenced by the inner branch of the Labrador Current, with lesser input from the Slopewater Current, a minor bifurcation from the Gulf Stream (Catto et al., 1999) (Fig. 1). The cold, inner Labrador Current flows south from Baffin Bay as a surface current and includes substantial outflow from Hudson Strait (Drinkwater, 1996). In contrast, the Slopewater current branches north from the Gulf Stream and brings warm, saline waters from the sub-tropics, at subsurface

depths, towards southern Newfoundland.

During the Last Glacial Maximum (~Marine Isotope Stage 2), Placentia Bay was glaciated by the Laurentide ice sheet; as a result, drumlins, moraines and megascale lineations are present across the sea floor (Shaw et al., 2006, 2013). The bay was deglaciated prior to the Younger Dryas climate reversal (12,800-11,600 cal yr BP; Pearce et al., 2013; Mangerud, 2021), although the precise timing of ice retreat is not well constrained (Dyke et al., 2002; Shaw et al., 2006, 2013). The varying sea

floor topography has resulted in heterogeneous deposition rates and differing basal ages are reported from sediment cores across the bay, which allows for the development of palaeo-records with various temporal lengths and resolutions. The potential for differing temporal records and sensitivity to elements of both the Labrador Current and Gulf Stream make Placentia Bay an ideal natural laboratory for studying past ocean/atmosphere interactions. As a result, numerous studies have developed palaeo-environmental records from the bay, all of which rely on radiocarbon chronology, necessitating the

adoption of marine reservoir corrections (e.g., Jessen et al., 2011; Solignac et al., 2011; Pearce et al., 2014; Sheldon et al., 2016).

## 2 Methods and materials

### 2.1 Core AI07-10G

Core AI07-10G measured 460 cm in length and was taken from 231.3 m water depth at 47.2389°N, 54.6140°W, in Placentia

Bay, North Atlantic Ocean. Sheldon et al. (2016) presented the results from radiocarbon dating (Table 1), Itrax-XRF core scanning, and benthic foraminiferal analyses. These results were combined with analyses from two other cores in Placentia Bay (12G and 14G; Sheldon et al. 2016) to form a composite full Holocene record.



## 2.2 Tephra extraction and analysis

To quantify volcanic glass shard concentrations in core AI07-10G (reported as shards per gram of dried sediment), we
processed continuous 5-cm wide samples taken throughout the sequence, with no gaps between samples, to identify sediment
intervals where cryptotephra deposits might be found (ranger finder counts). We subsequently analysed the 5-cm intervals
where tephra grains were identified at 1-cm intervals to pinpoint the position of cryptotephra deposits (Pilcher and Hall,
1992).

We extracted glass shards from the host material by drying samples at 105 °C overnight before immersing the material in 10
% hydrochloric acid and sieving them at 80 µm and 25 µm. Larger size fractions (>80 µm) were retained; however, given
the low shard concentrations in core AI07-10G (<40 shards per gram) were not investigated further (Abbott et al., 2018a).
Following this, we used stepped, heavy liquid (sodium polytungstate) floatation at 2.00 g/cm$^3$ and 2.50 g/cm$^3$ to concentrate
volcanic glass, which was mounted on slides and counted under a high-power microscope (Turney et al., 1998).

As no basaltic glass (which is denser than rhyolitic glass) was observed in the initial counts, we extracted glass shards for
electron probe microanalysis (EPMA) by sieving samples at 20 µm, followed by heavy liquid floatation at 2.15 g/cm$^3$ and
2.45 g/cm$^3$. The extracted material was then mounted in epoxy resin within acrylic stubs and polished to expose the internal
glass surfaces before carbon coating (Lowe et al., 2011).

The chemical compositions of individual glass shards (one analysis each) from samples taken at 195-190 cm and 35-30 cm
were determined by EPMA, with wavelength dispersive spectrometry on a JEOL 8900 Superprobe at the University of
Alberta. A suite of 10 elements (Si, Ti, Al, Fe, Mn, Mg, Ca, Na, K, Cl) were measured using a 5 µm beam diameter with a
15 keV accelerating voltage, and 6 nA beam current, with time-dependent intensity corrections applied to Na to compensate
for the smaller beam diameter (e.g., Jensen et al., 2008, 2021). In addition, we ran two secondary standards of known
compositions alongside samples from Placentia Bay to check for instrumental drift and analytical precision: i) Lipari
rhyolitic obsidian ID3506 and ii) Old Crow tephra (Kuehn et al., 2011). The major-minor element compositions of glass
shards are presented as normalised weight percent (wt %) oxides in comparative diagrams. The complete dataset and
associated standard measurements are reported in the supplementary information (Tables S1, S2).

## 2.3 Bayesian age-depth modelling

To incorporate chronological information from the ultra-distal cryptotephra isochrons identified in core AI07-10G, we
developed two different Bayesian age-depth models using OxCal v 4.4.4 (Bronk Ramsey 2009). The full code for both
models is available in supplementary information (supplementary text 1.1).

Model I is conceptually the same as the age-depth models described by Sheldon et al. (2016). In this model, radiocarbon
dates (Table 1) were calibrated with the Marine20 curve (Heaton et al. 2020) with a single reservoir correction applied to the
whole core. In this case, we have informative preliminary information regarding the most probable reservoir correction range
based on near-modern radiocarbon dating of marine organisms. In Bayesian statistics, this data is called a prior, e.g., a





representation of the state of knowledge regarding a parameter, expressed as a probability distribution, before considering all available information (e.g., stratigraphic context). For Model I, we used a prior distribution for the reservoir correction of the weighted mean of the 20 nearest points from Reimer and Reimer's (2001) marine reservoir correction database (Table S3), updated for use with Marine20 (-29±45 yrs). The core top was also included as an age constraint. We assume an exponential prior at zero depth, from 2007.7 CE (the approximate date of collection) decaying to 1000 years earlier with a time constant

($\tau$) of 50 yrs. The deposition was modelled as a Poisson process (i.e., a P_Sequence; Bronk Ramsey, 2008) with a nominal number of depositional events ($k_0$) of 1 per cm. The k parameter was permitted to vary within a wide range (i.e., two orders of magnitude on either side of $k_0$) and was selected through Markov chain Monte Carlo (MCMC) iterations (Bronk Ramsey and Lee, 2013). These settings are the default for sequences with a depth scale in centimetres.

Model II is similar to Model I; radiocarbon dates are calibrated with the Marine20 curve, the same core top constraint is

applied, and the model is formulated as a P_Sequence using the same parameters. However, Model II differs from Model I in two ways: (1) it includes cryptotephra deposits to constrain the chronology further, and (2) the model calculates multiple ΔR values -varying the radiocarbon reservoir offset throughout the sequence. We used the Mazama Ash and White River Ash eastern lobe (WRAe) as age constraints, both of which are geochemically verified in core AI07-10G (see 3.1 Tephrostratigraphy). Shard counts from both cryptotephra deposits consist of low concentrations and do not have a clearly

defined, sharp peak (Fig. 2). These variable and broader peaks are likely caused by downward translocation of shards through sediment loading or bioturbation (Griggs et al., 2015), and complicate the precise stratigraphic depth of the isochrons. In order to incorporate this uncertainty into the Bayesian models we took a conservative approach and used age uncertainties associated with the 5-cm ranger finder counts, rather than the 1-cm point finder counts. To do this, we first estimated the sediment deposition rate from model I at the central depth of both tephra samples. Then, we propagated the

depth uncertainty to the tephra age by adding uniform noise in the time dimension. The prior for each cryptotephra deposit was modelled using ages derived from ice core layer counting (a normal distribution; Sigl et al., 2016, 2022; Toohey and Sigl, 2017) plus chronologic sampling uncertainty (u), where u = sampling resolution/deposition rate.

Model II also differed from Model I and earlier approaches (Solignac et al., 2011; Sheldon et al., 2016) by including multiple independent ΔR estimates (i.e., each radiocarbon date had its own ΔR estimate). Each ΔR value was defined (as above) by a

mean correction of -29, however, the uncertainty was expanded to ±224 years (i.e., 4x the uncertainty from Reimer and Reimer's 2001 database). This conservative uncertainty regime was adopted to permit the MCMC approach inherent to OxCal's sequence modelling to generate an appropriate (non-truncated) posterior estimate for the corrections. A normal -29±224 prior was assumed for all radiocarbon dates except one. For the upper-most radiocarbon date (AAR-15764) we provided an even more forgiving prior (a uniform distribution centred at zero and spanning 2000 years). This prior was

selected because the WRAe mean depth is only 2 cm above this sample (Fig. 2). The MCMC is kept flexible by giving a wide uniform prior. Therefore, the tephra age can strongly inform the ΔR for this date, providing a clearer picture of the necessary reservoir effect around WRAe time.



## 3 Results and discussion

### 3.1 Tephrostratigraphy

Using multiple lines of evidence, we identified two discreet cryptotephra deposits in core AI07-10G (Fig. 2) that can be robustly correlated with volcanic eruptions in North America. Evidence included stratigraphic order, shard morphology and glass major-minor elements (wt%), which were interrogated using both bi-plots, Principle Component Analysis (PCA) and similarity coefficients (Borchardt et al., 1972). In both cases, the glass EPMA data were consistent and did not include glass shards with different chemical compositions or signs of weathering that might indicate reworking (Abbott et al., 2018a).

**3.1.1 Cryptotephra deposit 10G_195 (Mazama Ash)**

Cryptotephra deposit 10G_195 is formed of colourless, platy and fluted shards, with rhyolitic chemical compositions (Fig. 2). Shard morphology, stratigraphy and glass major-minor elements (similarity coefficient 0.95) are all consistent with Mazama Ash (Fig. 3), which has been identified in study sites throughout north-eastern North America (Pyne-O'Donnell et al., 2012; Spano et al., 2017; Jensen et al., 2021). The Mazama Ash was derived from a VEI 7 (Volcanic Explosive Index)

eruption of Mount Mazama (Crater Lake), Oregon, that was amongst the largest volcanic eruptions to take place during the Holocene with an estimated erupted volume of ~176 km$^3$ (Buckland et al., 2020). Visible ash layers from this event extend throughout much of western North America (Jensen et al., 2019), and cryptotephra deposits are reported in the Greenland ice cores and, potentially, Western Europe (Zdanowicz et al., 1999; Plunket and Piltcher, 2018). Mazama Ash has been precisely dated to 7572±18 yr BP by ice core layer counting (Zdanowicz et al., 1999; Sigl et al., 2016, 2022) and 7682–7584 cal. yr

BP by Bayesian age modelling, including from 81 radiocarbon dates (Egan et al., 2015).

**3.1.2 Cryptotephra deposit 10G_35 (White River Ash eastern lobe)**

Cryptotephra deposit 10G_35 is composed of colourless, highly vesicular or pumiceous shards with rhyolitic chemical compositions (Fig. 2). Shard morphology, stratigraphy and glass major-minor elements (similarity coefficient 0.95) are all consistent with White River Ash eastern lobe (WRAe) (Fig. 3), which has been identified in study sites throughout north-

eastern North America (Pyne-O'Donnell et al., 2012; Mackay et al., 2016, 2022; Monteath et al., 2019; Jensen et al., 2021). The WRAe is derived from a magnitude 6.7 (VEI 6; erupted volume 39.4-61.9 km$^3$) Plinian eruption of Mt. Churchill, Alaska (Lerbekmo, 2008; Mackay et al., 2022), and extends eastward from the Wrangell Volcanic Field. Ash from this eruption has been identified in the Greenland ice cores and numerous study sites from western Europe, where it was first described as the AD860 cryptotephra (Coulter et al., 2012; Jensen et al., 2014). The WRAe has been precisely dated by ice

core layer counting, which constrains the eruption timing to the winter of 852/853±1 C.E. (Toohey and Sigl, 2017) – consistent with proximal stratigraphy and Bayesian age modelling (using 28 radiocarbon dates) that dates the eruption to 1175-1075 cal yr BP (West and Donaldson, 2000; Davies et al., 2016).





### 3.2 Bayesian age-depth modelling

The presence of Mazama Ash and WRAe allows for the marine reservoir offset to be assessed at multiple points during the

Holocene. Results from Model II, which includes the cryptotephra isochrons, show that around Mazama Ash the radiocarbon offset was moderately more negative (i.e., -126±151 relative to the prior ΔR of -29±45 years) (Fig. 4). The large uncertainty range (relative to the offset) associated with the Mazama Ash is caused by our conservative modelling approach that uses the 5 cm range finder results to place the isochron, and the slow accumulation rate at this point in the core (~0.05 cm yr$^{-1}$). Around WRAe, the radiocarbon offset was larger, e.g., -396±144 (Fig. 4). Around the ages of both tephra deposits, ΔR

values must be more negative than previously assumed to account for the tephra ages. That is, modelled ages are made to be older than would be suggested by the original prior, therefore generally less old carbon is contributing to the system at the study site than modelled for the global ocean. The ΔR varies considerably throughout the age-depth model and particularly around the WRAe isochron. At this depth, there is a large shift in ΔR near WRAe. We modelled a posterior offset of -451±151 years for radiocarbon date AAR-15764, but only -10±213 years for radiocarbon date AAR-17060, 42 cm lower in

the core. Therefore, although the reservoir age for both periods of tephra deposition was lower than indicated by Reimer and Reimer's (2001) marine reservoir correction database, it was substantially lower in the Early Holocene than in the Mid Holocene. This large difference in ΔR may be explained by either artefacts in the chronology (e.g., radiocarbon date AAR-15764 is inaccurate) or real variance in the age of water bodies. However, as discussed below, the two tephra isochrons were deposited during periods characterised by different hydrographical conditions, and the difference in ΔR for the two tephra

deposits likely reflects real differences in the radiocarbon age of the water bodies affecting the site.

Across the whole core, there is an average ΔR difference of 74 years, but as high as 416 yrs at 34.5 cm, not far below the WRAe, and 126 (a secondary maximum) directly at the mean Mazama depth. In most places, these changes do not represent a departure beyond the two-sigma age range of Model I. Indeed, the only portion of the core that does exceed this range and precludes the implicit null hypothesis (no change) is near the WRAe (31.8-35.7 cm) (Fig. 4). Both cryptotephra isochrons

push the Bayesian model towards older values (Fig 4.). It is possible that this is caused by inaccurate placing of the position of the cryptotephra isochrons and that Model I is correct. For this to be the case then both cryptotephra isochrons would be expected to occur deeper in the core (e.g., WRAe would have had to occur at 39.3 cm depth – almost 5 cm below the observed peak at 35-30 cm depth; Fig. 2) and the observed peak in shard abundance would need to have been reworked upwards into the overlying sediments. Upward movement of the cryptotephra deposits to an extent where the position of the

isochron is misplaced seems unlikely, however, as in both cases shard counts are considerably higher at the denoted isochron depth (which already includes 5 cm uncertainty) than below.

Considering the marked departure of ΔR around the time of the precisely dated WRAe from the near-modern prior, we observe that radiocarbon reservoir effects can shift rapidly because of environmental and systemic changes (e.g., carbon source and ocean circulation shifts) over time. Further, the reservoir correction uncertainty may be more substantial in

marine settings than suggested by near-modern samples. We conclude that conventional means of relaying proxy records




over time often fail to account for time uncertainty. A natural remedy to this failure, and one we advocate for palaeoenvironmental proxy studies, is to propagate the age-uncertainty of an age model ensemble to proxy records (e.g., McKay et al., 2021).

## 3.3 The implications, potential and challenges of using ultra-distal tephra isochrons in ocean-sediments

### 3.3.1 Implications and potential applications

Comparative tephrochronological and radiocarbon-dated age models have provided good evidence for past changes in water masses (Knudsen and Eiríksson, 2002; Eiríksson et al., 2004, 2011) and we revised the AI07-10G core chronology with similar aims to improving understanding of regional ocean circulation. Sheldon et al. (2016) suggest that the influx of the warm Slopewater Current dominated the area in the Early-Mid Holocene, when the Mazama Ash (7622±18 C.E.) was
deposited. After ca. 7300 cal yr BP, the Labrador Current strengthened, weakening the inflow of the warmer Slopewaters. Even though the Labrador Current weakened again in the Late Holocene (after ca. 4000 cal yr BP), during which the WRAe Ash (852-853 C.E.) was deposited, the influence of the Slopewater Current did not become as pronounced as in the Early Holocene. Therefore, the difference in ΔR seen at the Mazama Ash compared with the WRAe Ash may reflect actual differences in the radiocarbon age of the water masses affecting Placentia Bay. It also suggests stronger ventilation, and
therefore younger reservoir age, of the Slopewater compared with the waters from the Labrador Current.

Identification of Mazama Ash and WRAe in ocean sediments from the north-western North Atlantic highlights the potential for using ultra-distal cryptotephra deposits to constrain marine radiocarbon offsets in this region. More than thirty unique glass populations have been identified in north-eastern North America (Jensen et al., 2021), many of which are correlated with eruptions with well-constrained (decadal or even annual) age ranges. Several of these provide opportunities to
synchronise marine records for differing ocean basins. For example, Aniakchak CFE II tephra is present in both the Chukchi Sea and the North Atlantic Ocean (Jennings et al., 2011; Pearce et al., 2017). In addition, other eruptions with less precise age constraints are routinely dated using Bayesian models to integrate large volumes of differing chronological data (e.g. Blockley et al., 2008; Keuhn et al., 2009; Davies et al., 2016). Combined with methodological advances in shard extraction (e.g. Turney et al., 1998; Blockley et al., 2005) and EPMA (e.g. Hayward, 2012), these techniques will no doubt continue to
enhance the power of tephrochronology and provide new opportunities to use this technique in marine settings.

### 3.3.2 Methodological and taphonomic challenges

Previous studies have identified numerous tephra and cryptotephra deposits in ocean sediment cores from the North Atlantic as part of a tephra framework founded on Icelandic eruptions (e.g., Abbott et al., 2018b). These studies have described several methodological and taphonomic complications that must be considered in our interpretations of cryptotephra deposits
from Placentia Bay and by future investigations of ultra-distal, North American cryptotephra deposits in ocean-sediments:





(i) Extracting sufficient shards for EPMA from low-concentration cryptotephra deposits is challenging, and the number of successful analyses is typically lower than shard counts. In ocean sediments, this is complicated by dominate silt (63-2 µm) and clay (<2 µm) size fractions that can be difficult to remove with sieving, as well as abundant biogenic silica that includes densities similar to glass. In this study, we used large sample volumes (>3 cm$^3$) and a narrow range of densities (2.15 g/cm$^3$
and 2.45 g/cm$^3$) during heavy liquid separation for EPMA to mitigate these complications. While we achieved successful results with this method, cryptotephra deposits include a diverse range of volcanic glass (morphological and chemical composition), and our approach may not be suitable in all settings. For example, heavy liquid densities of ≤2.45 g/cm$^3$ are unsuitable for extracting denser basaltic glass from host sediments.

(ii) Separating primary air fall events from reworked or ice-rafted detrital glass is a challenge in large parts of the North
Atlantic that are affected (both directly and indirectly) by Icelandic volcanism (Abbott et al., 2018a). In this respect, settings such as Placentia Bay, which is sheltered from the strongest ocean currents and largely unaffected by ice rafting, may be more suitable for preserving discreet tephra isochrons. The low shard concentrations in our study (<40 shards per gram) highlight the importance of site location and the sensitivity of ultra-distal cryptotephra deposits to background noise that could easily obscure the isochrons. A second example of identifying low-concentration cryptotephra deposits in the North
Atlantic is provided by Jennings et al. (2014). They report the presence of Aniakchak CFE II in an ocean core taken immediately east of Greenland. The coring site lies within the East Greenland Current that brings polar, and importantly tephra-free, waters south – reaffirming the importance of site location and ocean conditions in successful studies.

(iii) Identifying the precise position of tephra isochrons in core AI07-10G is difficult as the peak in shard counts is not obvious in either deposit - both of which are composed of low shard concentrations without clear, discrete peaks above
background noise. These complications are common in cryptotephra deposits (Lowe, 2011; Davies, 2015; references therein) in ocean sediments and can be exacerbated by bioturbation or sediment loading (Griggs et al., 2015). Because of these limitations, we suggest a conservative approach when using ocean cryptotephra deposits to synchronise palaeoenvironmental records (as we did) if isochrons are not clearly resolved in shard counts. Future studies, however, may identify better-resolved isochrons and there is potential to develop marine-terrestrial-cryosphere linkages using ultra-distal cryptotephra
deposits.

## 4. Conclusions

Tephrochronology provides a means to establish local marine radiocarbon offsets. Understanding these offsets is essential in developing a robust chronology for ocean palaeoenvironmental records. In this study, we identify the Mazama Ash and White River Ash eastern lobe (WRAe) in Placentia Bay, North Atlantic Ocean. The precise ages of these isochrons and
occurrence in depths close to radiocarbon dates allow us to refine the local marine radiocarbon reservoir to -126±151 years at ca. 7572±18 yr BP (the age of Mazama Ash) and -396±144 years at ca. 853±1 C.E. (the age of WRAe). Changes in ΔR coincide with inferred shifts in water masses. The smaller absolute value of ΔR at the time of Mazama ash deposition occurs

during a period, when the Slopewater Current is suggested to have strongly affected the Placentia Bay. The larger, more
negative ΔR at the time WRAe deposition took place during a period when the Labrador Current was more influential
(although still not dominant). By incorporating these chronological data within a Bayesian age-depth model with a variable
radiocarbon offset (ΔR) we develop a chronology that better reflects uncertainties regarding marine carbon. Our findings
demonstrate that reservoir ages may vary substantially within the Holocene. Therefore, it is critical to consider potentially
variable ΔR when ocean circulation and ventilation characteristics have differed over time. Results from this study, and
others in the North Atlantic, indicate that site location is an important factor in the preservation of marine cryptotephra
isochrons, which are strongly impacted by taphonomy and ice rafting. Therefore, sheltered bays or areas influenced by
currents that are unlikely to include volcanic ash are preferable.

## Author contributions

AJM conceived the project. AJM, JH & BJ undertook cryptotephra analysis and interpretation. MSMB undertook Bayesian
age modelling. MSS and CP provided chronology and materials. AJM wrote the manuscript with input from all the authors.

## Acknowledgements

Are we are grateful to a Natural Sciences and Engineering Research Council of Canada Discovery Grant awarded to B.
Jensen. We also acknowledge the Danish Council for Independent Research (grant no. 0135-00165B (GreenShelf) to MSS);
the project has also received funding from the European Union's Horizon 2020 research and innovation program under
Grant Agreement No. 869383 (ECOTIP) (MSS).

## Competing interests

Dr Britta Jensen is a member of the editorial board of Geochronology.

## Code/data availability

All code/data used in this project is made available in Suplementary Information.

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





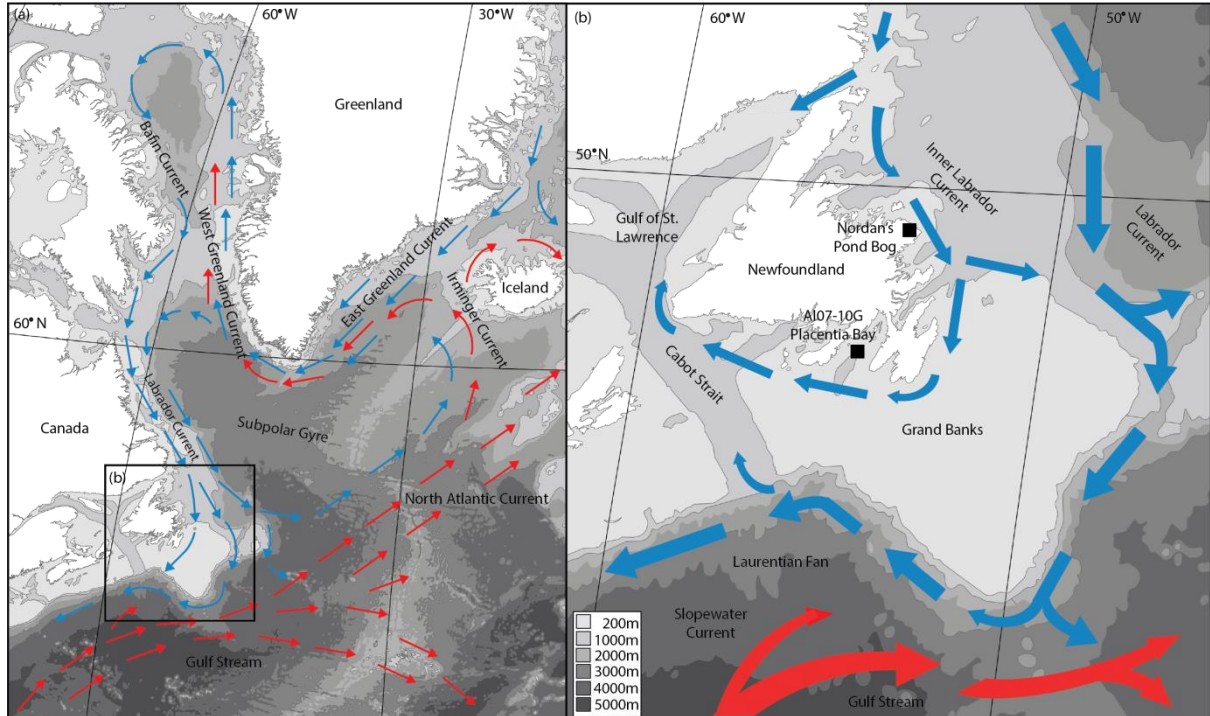

**Figure 1: (a) Map showing the major surface/subsurface currents in the North Atlantic. (b) The surface and subsurface currents affecting Newfoundland and Placentia Bay. In both maps, blue arrows indicate cold, polar water, while red arrows indicate warmer, Atlantic-sourced water.**



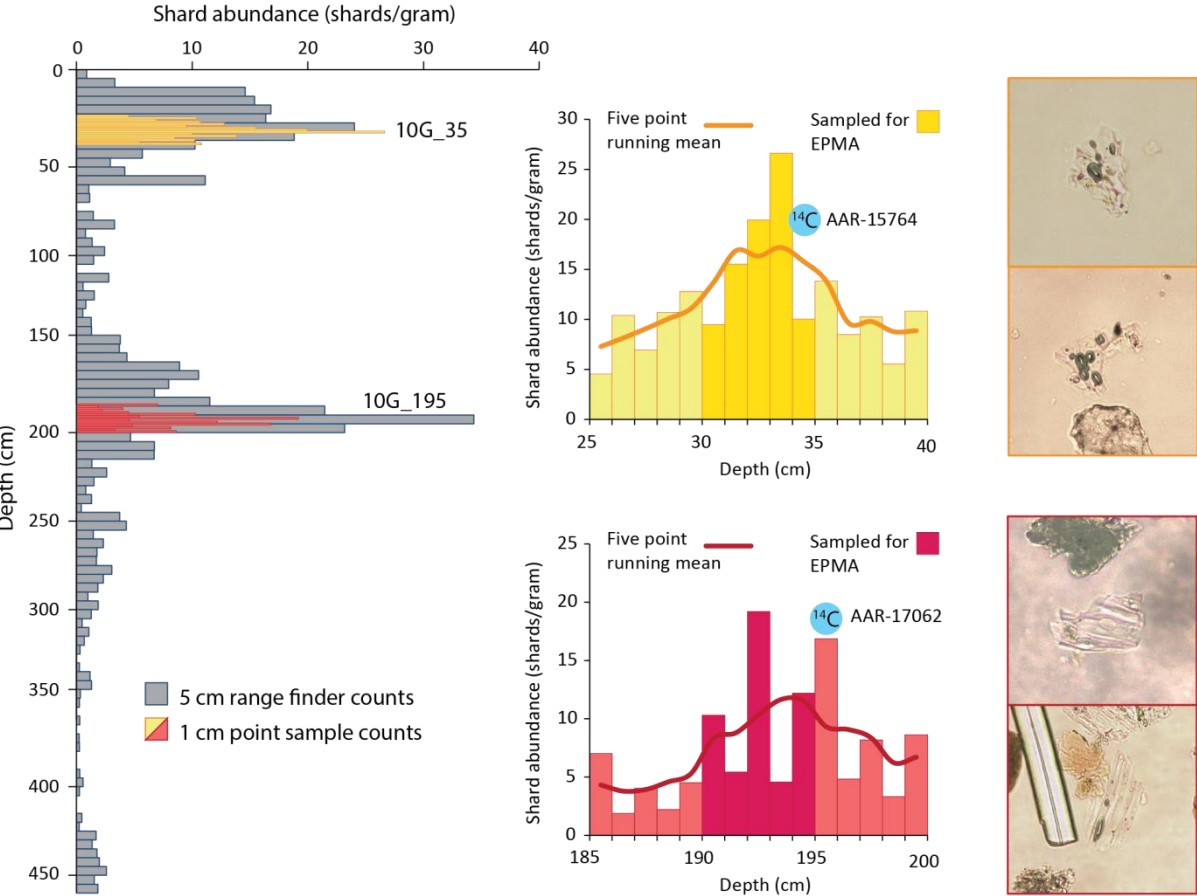

**Figure 2: Shard counts and images from core AI07-10G, Placentia Bay, North Atlantic Ocean. Two shard peaks were found**
**centred around core depths 32.5 cm (cryptotephra deposit 10G_35) and 192.5 cm (cryptotephra deposit 10G_195).**



**Figure 3: (a-c) Bi-plots of glass major-minor elements. (b) Principle Component Analysis scores derived from glass major-minor elements. Comparative data includes EPMA analyses of Mazama Ash, excluding dacite shards which are rarely present in north-eastern North America (Jensen et al., 2019), and White River Ash eastern lobe (WRAe) (Jensen et al., 2014). Note that the three outlying 10G_195 analyses in panel c all have low analytical totals (<95%) (Table S1) and elevated Cl, which is likely to be derived from the epoxy resin mounting agent.**




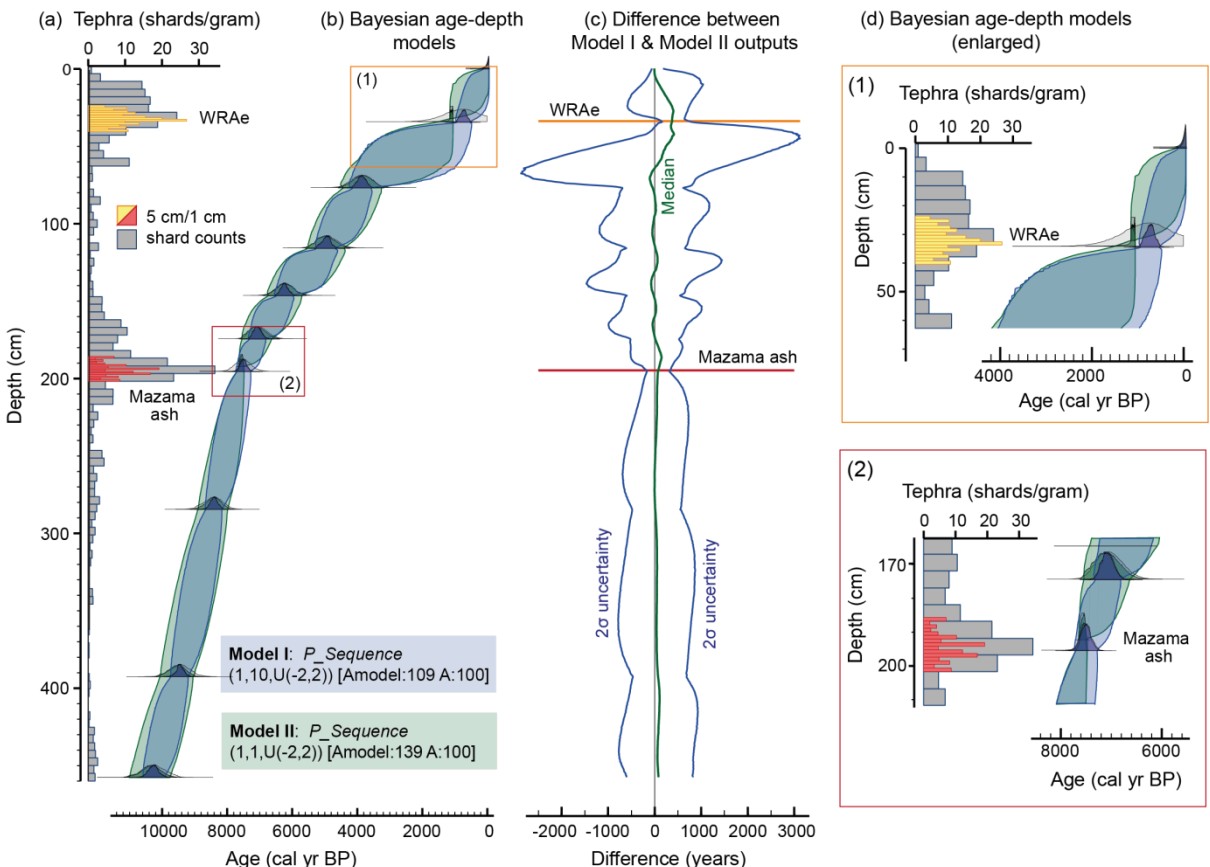

**Figure 4: (a) Shard counts from core AI07-10G, Placentia Bay, North Atlantic Ocean. (b) Oxcal P_Sequence age-depth models. (c)**
**The difference between Model I and Model II outputs. (d) Oxcal P_Sequence age-depth models zoomed in around the WRAe and**
**Mazama Ash. All Oxcal models are shown at two sigma (95.4%) uncertainty. Light grey probability density functions show prior**
**likelihoods; dark grey are posterior likelihoods.**

**Table 1: Radiocarbon ($^{14}$C ) dates from core AI07-10G, Placentia Bay (Sheldon et al., 2016). Modelled age (I) refers to Bayesian**
**age-depth model I and uses a single ΔR of -29±45 years. Modelled age (II) refers to Bayesian age-depth model II and uses a**
**variable ΔR between -29±224 years. The ΔR is reported as mean and one standard deviations as this is routine for such data,**
**making it easier to include in future age-depth modelling efforts.**

| Lab n. | Depth (cm) | Material | $^{14}$C age | Calibrated age (cal yr BP) | Modelled age (I) (cal yr BP) | Modelled age (II) cal yr BP) | ΔR (Model II) |
|---|---|---|---|---|---|---|---|
| AAR-15764 | 34-35 | Mixed benthic foraminifera | 1306±70 | 886-540 | 938-535 | 1491-1051 | -451±151 |
| AAR-17060 | 76.5-77.5 | Mixed benthic foraminifera | 3993±66 | 4060-3595 | 4145-3574 | 4420-3247 | -10±213 |
| AAR-15765 | 115.5-116.5 | Mixed benthic foraminifera | 4821±67 | 5177-4665 | 5259-4683 | 5525-4407 | -50±210 |
| AAR-17061 | 146-147 | Mixed benthic foraminifera | 5979±70 | 6399-5984 | 6483-5987 | 6738-5710 | -25±221 |
| AAR-15766 | 174-175 | Mixed benthic foraminifera | 6730±69 | 7246-6821 | 7318-6818 | 7552-6633 | -50±210 |
| AAR-17062 | 195-196 | Mixed benthic foraminifera | 7199±73 | 7669-7310 | 7709-7278 | 7706-7491 | -91±106 |
| AAR-15767 | 284-285 | Gastropod (*Nuculana minuta*) | 8072±73 | 8576-8171 | 8655-8160 | 8901-7994 | -50±191 |
| AAR-15768 | 392-393 | Gastropod (*Nuculana minuta*) | 8905±70 | 9600-9185 | 9736-9200 | 10086-9064 | -103±201 |
| AAR-12117 | 456-459 | Mixed benthic foraminifera | 9521±86 | 10495-9930 | 10583-9919 | 11027-9778 | -101±209 |