# Peer review of "Ultra-distal tephra deposits and Bayesian modelling constrain a variable marine radiocarbon offset in Placentia Bay, Newfoundland"

_EGUsphere, 2022_

## Author Response (AR1)

Dr. A.J. Monteath
Geography and Environmental Science
University of Southampton
Southampton, Hampshire, SO171BJ
Tel: +44 (0)794 177 8714
Email: a.j.monteath@soton.ac.uk

[Figure]

March 23rd, 2023

**RE: Submission of manuscript for consideration**

Dear Geochronology Editorial Staff,

We re-submit our manuscript entitled; ***Ultra-distal tephra deposits and Bayesian modelling constrain a variable marine radiocarbon offset in Placentia Bay, Newfoundland***, for your consideration as a research article in Geochronology. The manuscript has been modified following the recommendations from the two reviewers. In both cases the comments were minor and did not require moderate or major changes.

We are very grateful to both reviewers for taking the time to carefully review our manuscript. To address the comments made by each reviewer we have provided a response to each individual comment below.

Yours sincerely,

Dr. Alistair Monteath

Geography and Environmental Science
University of Southampton
Southampton, Hampshire, SO171BJ
Email: a.j.monteath@soton.ac.uk

Mr. Matthew Bolton

Earth and Atmospheric Sciences
University of Alberta
Edmonton, Alberta, T6G 2E3
Email: bolton1@ualberta.ca

Reviewer One

"Monteath and co-authors present compositional data from two well-dated North American cryptotephra deposits in a marine sediment core from Placentia Bay, Newfoundland. By incorporating the ages of these tephra layers with previous radiocarbon dates from the same sediment core, and application of Bayesian age modeling, the authors make inferences about the past variability of the local marine radiocarbon offset (deltaR) through the Holocene. I found the paper to be well-written with clear accompanying figures and datasets. The methods are strong, particularly the Bayesian statistics, although I have a suggestion related to PCA analyses using compositional major oxide data (see L162-163 below). Beyond this, I only have minor suggestions and think that a revised manuscript would be appropriate for publication in Gchron. I look forward to seeing future applications of cryptotephra layers in marine sediment records to constrain variable deltaR like this!"

L18: CE should be BP for Mazama Ash age.

*'C.E'. has been corrected to 'yr BP'.*

L20: 'Current' is misspelled.

*'Current' has been corrected.*

L36: It would be helpful for non-14C specialists to know what range of average reservoir ages are here for context.

We added '*During the Holocene, the global average marine reservoir age varies between 700-350 years (Heaton et al., 2023)'.*

L91: Since you refer to this data later on in the discussion, please clarify what foram analyses were performed. Assemblages, isotopes, etc?

We '*changed foraminiferal analyses*' to '*foraminiferal assemblage analyses*'

L122: Please specify what this single reservoir correction is here for reference.

We added '*(-29±45 yrs)'*

L160 and 262: Discrete is misspelled.

*'Discrete' has been corrected.*

L162-163: (i) Have the authors considered the use of log-transformation ratios to better separate tephra sources? Especially when it comes to statistical analyses like PCA, these analyses cannot be performed on compositional oxide data due to the constant sum constraint, i.e., the co-dependency of variables. In other words, if one variable changes, they all do as they are fractional abundances. (ii) In addition, it would be helpful to have more information on the similarity coefficient analyses performed. (iii) Any references for the software, R packages, etc that were used would be important to include here as well.

(i) We appreciate the reviewer's important point regarding log-ratio transforms for compositional data, particularly when considering PCA. In our original submission, we conducted PCA on closed data using the "conventional" approach, which is a well-known procedure but has issues the reviewer correctly identified. This conventional method and its shortcomings for closed data have been

discussed extensively, at least as far back as Aitchison (1983), and addressed by Chayes before that in the 60s.

We acknowledge that Compositional Data Analysis is gaining traction in the tephra field. We agree with the reviewer that taking advantage of modern compositional analysis techniques would be the proper approach. Thus, we plan to incorporate log-ratio transforms into the revised work and re-run PCA on our compositional data using these methods (though, for clarity, aside from PCA, keeping standard, normalized [not log-ratio] composition plots). We believe this will help us clarify the relationships between the variables while still achieving the dimensionality reduction task of PCA.

Considering this, we will apply the "pcaCoDa" function from the "robCompositions" R package (Templ et al. 2011), running a "compositional" version of traditional PCA on the data, following the method outlined by Filmoser et al. (2009). We will first convert the data into isometric log-ratio coordinates and then apply classical PCA to the transformed data. Finally, we will retransform the resulting loadings and scores to centred log-ratio space to produce a compositional biplot.

(ii) We added the formula used to calculate the similarity coefficient analyses to the supplementary information and made reference to this in the main manuscript.

(iii) Packages and sources for these methods include the following and will be referenced in the revised manuscript.

Filzmoser, P., Hron, K., Reimann, C. (2009) Principal component analysis for compositional data with outliers. Environmetrics, 20, 621-632.

Matthias Templ, Karel Hron, Peter Filzmoser (2011). robCompositions: an R-package for robust statistical analysis of compositional data. In V. Pawlowsky-Glahn and A. Buccianti, editors, Compositional Data Analysis. Theory and Applications, pp. 341-355, John Wiley & Sons, Chichester (UK).

Peter Filzmoser, Karel Hron, Matthias Templ (2018). Applied Compositional Data Analysis. With Worked Examples in R. Springer Series in Statistics. Springer International Publishing, Cham, Switzerland.

L173-175: Which Mazama Ash date was used for the age model?

We amended line 137 to clarify this for the reader: *'We used Mazama Ash (7572±18 yr BP; Sigl et al., 2016, 2022) and White River Ash eastern lobe (WRAe) (852/853±1 C.E.; Toohey and Sigl, 2017) as age constraints, both of which are geochemically verified in core AI07-10G'*.

L184-187: Which WRAe date was used for the age model?

We amended line 137 to clarify this for the reader: *'We used Mazama Ash (7572±18 yr BP; Sigl et al., 2016, 2022) and White River Ash eastern lobe (WRAe) (852/853±1 C.E.; Toohey and Sigl, 2017) as age constraints, both of which are geochemically verified in core AI07-10G'*.

L199: Please provide depth of 14C layers in core and/or uncorrected age here for context. Also please include the depth for the WRAe tephra layer. Maybe just put in paratheses following the sample name.

All data regarding the radiocarbon dating is listed in table 1. We added *'was identified between 35-30 cm depth (32.5 cm depth peak)'* to line 184.

L201: I wouldn't use the word 'substantially' as it is rather subjective and the deltaR uncertainty overlap between Early and Middle Holocene, making them not statistically different. Also please provide the general difference for reference.

We removed the word '*substantially*'.

L200-203: This sentence seems out of place as the discussion before was focused on the deltaR around the WRAe, and then you led this sentence with 'therefore'. Consider rephrasing for clarity. Or do you mean it was lower in the Late Holocene compared to the Early-Mid Holocene?

We have re-drafted this sentence to read: '*The reservoir age for both periods of tephra deposition was lower than indicated by Reimer and Reimer's (2001) marine reservoir correction database, however, this offset appears larger in the Early Holocene than in the Mid Holocene*'.

L228: It would be helpful if you provided what sort of proxy evidence these paleoceanographic changes are based on for reference.

We amended line 228 to clarify this for the reader: '*Sheldon et al. (2016) used Itrax-XRF core scanning, and benthic foraminiferal assemblage analyses to suggest that the influx of the warm Slopewater Current dominated the area in the Early-Mid Holocene, when the Mazama Ash (7572±18 yr BP) was deposited*'.

L264: (i) In regard to background noise, could you try change-point analyses, or something similar, to objectively determine when concentrations rise substantially? (ii) As I am not a crypto specialist, what is considered background versus 'primary' airfall? (iii) Since there are tephra shards always present in your record, at what threshold do you decide to increase sampling resolution for EMPA analysis? Some sort of objective statistical test would be nice to see applied. (iv) Additionally, what do you suspect is the source of constant tephra deposition if Placentia Bay is presumably sheltered oceanographically

(i) As tephra deposition is typically non-continuous, to our knowledge change-point analysis has not previously been undertaken on cryptotephra records. As the Placentia Bay record contained so little tephra we sampled every core area where we thought it may be possible to extract sufficient glass for EPMA following established sampling protocols (Pilcher and Hall, 1992). It would be very interesting to attempt change-point analysis on records with more abundant tephra concentrations, however.

(ii) Distinguishing between background noise and primary airfall events is a difficult (and often subjective) task that will vary from site to site. In proximal locations background noise may be characterised by many thousands of shards, whereas, in ultra-distal locations background noise can be a little as a few shards, or even entirely absent. In ocean sediments bioturbation and ice-rafting can make this task still more difficult. For the Placentia Bay record, we essentially assumed that all tephra shards were derived from background noise unless they met conditions outlines by Abbott et al. (2018). These consistent results are briefly described in section 3.1.

(iii) We simply increased the sampling resolution around the two highest shard concentrations. Further core sections could be targeted for higher sampling resolution, however, in this record we are already working at the practical limit for cryptotephra analyses, and is unlikely that it would be possible to extract sufficient shards for EPMA.

(iv) The continual background of tephra throughout the Placentia Bay record could come from a number of different sources, including; loess deposits in the mid-west (Jensen et al., 2021), ice-rafting

(from the limited bergs that do enter the bay), eruptions taking place in the Northern Hemisphere or reworking of primary cryptotephra deposits present in the core. Background tephra deposition is a common occurrence in most sedimentary environments. In the Placentia Bay record this noise is more obvious because the peaks in shard-abundance are so subtle. For example, the WRAe peak in Nuangola Lake (northwest Pennsylvania) is quantified at >16,000 shards per gram (Monteath et al., unpublished data), whereas, in Placentia Bay the WRAe is quantified as 34 shards per gram.

L267: But the East Greenland Current carries substantial amounts of sea ice. Are you sure no tephra shards are deposited in the sea ice source areas?

We amended line 267 to clarify our point to the reader: '*The coring site lies within the East Greenland Current which brings polar waters that are less affected by tephra-ice rafting from Iceland, south – reaffirming the importance of site location and ocean conditions in successful studies*'.

Reviewer Two

"This manuscript concisely reports new geochemical and chronological data from sediment cores off the coast of Newfoundland. The authors successfully utilize tephra correlations to improve their age-depth model and to better understand local marine radiocarbon reservoir variability over time. I find the research fits the aims of Geochronology, and the methods used are appropriate for the aims of the study. The results will be of interest for those working on stratigraphic and chronological correlations of Northern hemisphere paleoclimate records. I recommend publication of the paper with minor revisions."

Line 13: change to "these isochrons *had* never been identified"

We have amended line 13 to: '*Previously, however, these isochrons have not been identified in sediments from the north-west Atlantic Ocean*'.

Line 17: Mazama ash age should be cal yr BP

'*C.E.*' has been corrected to '*yr BP*'.

Line 19: In my opinion the wording of the sentence "Changes in $\Delta R$ coincide with inferred shifts in relative influences of the Labrador Current and the Slopewater curret in the bay" is stronger than justified by the evidence. This is a likely cause of changes in $\Delta R$, but other causes remain possible, and the timing (coincide) of changes in $\Delta R$ is not well constrained.

We changed '*Changes in $\Delta R$ coincide with inferred shifts…* to *Changes in $\Delta R$ appear to coincide with inferred shifts…*'

Line 89: Explicitly write the year of coring in this paragraph.

We added '*drilled in 2007*' to the paragraph.

Line 100-101: check grammar for clarity

We changed '*host material* to *host sediment*' to remove repetition of '*material*' in this sentence.

Line 113: How small is "smaller"? Or if 5 um is already smaller, then clarify by saying what it is "smaller" relative to.

*We changed 'smaller' to 'narrow beam (<10 μm)'.*

Line 185: Please write both dates in the same scale so they can easily be compared.

*We have changed 'C.E.' to 'yr BP' throughout the manuscript for consistency.*

Line 229: Mazama age to cal yr BP

*'C.E.' has been corrected to 'yr BP'.*

Line 231-235: You say that the Slopewater current was not as pronounced around the time of the WRAe compared to the early Holocene. But the very negative ΔR implies much younger water masses. Isn't this contradictory? Is the claim that the Slopewater current was not as strong around 850 C.E. as during the early Holocene from Sheldon et al., 2016? Then clarify that. And maybe state what proxy or method that interpretation is based on. Clarify that your data show a different pattern for this time period than their study, and mention possible reasons for the discrepancy. Have other studies provided information about the relative ages of these waters? If yes, cite. If no, this may be emphasized as an interesting new finding.

*We have added 'Sheldon et al., 2016) to line 234 to emphasise that evidence for changes in the strength of the Slopewater Current comes from this study.*

*We agree that line 235 is confusing and should be clarified. The cause of changes in the radiocarbon age of waters in Placentia Bay are not clear cut and could be due to a number of factors (e.g., changes in the relative strength of ocean currents, stronger ventilation or increasing terrestrial run off). We have amended the manuscript to read: 'Therefore, the difference in ΔR seen at Mazama Ash compared with the WRAe Ash may reflect actual differences in the radiocarbon age of the water masses affecting Placentia Bay. It also suggests the inner Labrador Current, which includes a substantial terrestrial component from Hudson Strait, has a younger reservoir age compared with the waters from the Slopewater Current.'*